# Validation of a Nursing Workload Measurement Scale, Based on the Classification of Nursing Interventions, for Adult Hospitalization Units

**DOI:** 10.3390/ijerph192315528

**Published:** 2022-11-23

**Authors:** María Fuensanta Hellín Gil, María Dolores Roldán Valcárcel, Ana Myriam Seva Llor, Francisco Javier Ibáñez-López, Marzena Mikla, María José López Montesinos

**Affiliations:** 1Faculty of Nursing, University of Murcia, 30100 Murcia, Spain; 2Biomedical Research Institute of Murcia (IMIB), 30120 Murcia, Spain; 3Education Faculty, University of Murcia, 30100 Murcia, Spain

**Keywords:** staff workload, nursing staff, hospital, nursing administration research, nursing care management, nursing

## Abstract

We conducted validation of a scale to measure nursing workloads, previously designed using NIC interventions within the four nursing functions (patient care, teaching, management, and research). Methods: This is an analytical, descriptive, prospective, and observational study using qualitative methodology (focus groups and in-depth interviews) with a quantitative and qualitative section (committee of experts and real application of the scale through a validation pilot and with multicentric application, including hospitalization units of internal medicine and surgery of four hospitals). Qualitative analysis was performed with Atlas.ti8 and quantitative analysis with R. Results: Qualitatively, all the participants agreed on the need to measure workloads in all nursing functions with standardized terminology. The expert committee found greater relevance (91.67%) in “prevention” and “health education” as well as consistency with the construct and adequate wording in 99% of the selected items. In the pilot test and multicenter application, the nurses spent more time on the caring dimension, in the morning shift, and on the items “self-care”, “medication”, “health education”, “care of invasive procedures”, “wounds care”, “comfort”, and “fluid therapy”. Cronbach’s alpha 0.727, composite reliability 0.685, AVE 0.099, and omega coefficient 0.704 were all acceptable. Construct validity: KMO 0.5 and Bartlett’s test were significant. Conclusions: The scale can be considered valid to measure nursing workloads, both qualitatively in obtaining the consensus of experts and health personnel and quantitatively, with acceptable reliability and validity superior to other similar scales.

## 1. Introduction

As recently demonstrated during the COVID-19 pandemic, the management of nursing human resources is a key aspect of quality in healthcare since nurses are significantly the largest group in any healthcare institution [1]. In health services, nursing accounts for approximately 70% of salaries and wages paid by health services budgets, and evidence as to the efficacy and effectiveness of any staffing methodology is required since it has workforce and industrial relationship implications [2]. Significant association has been demonstrated between workloads in hospitalization units and the average length of hospital stay, increased morbidity, mortality, and patient satisfaction [3]. This confirms that increased workload has an impact on the quality of health care and patient safety, and therefore, adequate staffing to care demands promotes safer care environments [3,4,5]. An adequate nurse–patient ratio promotes a safer care environment, increasing the quality of care and patient safety while allowing the achievement of professional objectives related to a job-satisfaction atmosphere [3,4,6,7]. Therefore, identifying and measuring adequately the workload of nursing professionals is an important indicator to achieving efficient and effective management of human resources and establishing the number of staff needed for proper health care, based on scientific evidence [8]. This situation of nurses is a worldwide problem in all specialties, but this is exacerbated in medical-surgical areas. In this field, high patient–nurse ratios, use of point-of-care technologies, and stressful working conditions require sufficiently highly skilled nurses, with few research studies available on perceived workload, burnout, and intention to quit among medical-surgical nurses [7,9].

Traditionally, attempts have been made to adjust the nursing staff by adapting patient–nurse ratios based on traditional parameters with little or no scientific basis or focusing only on the patient’s level of dependence [4], except in critical care units, where there are precedents in the literature for the creation of validated measurement instruments for nursing workloads [10]. These instruments are usually designed based on the characteristics of these units and their patients, and although some attempts at application in hospitalization units have been made [11], they are not suited to the actual circumstances and demands of patients admitted to these units, and there is a significant research gap on methodologies for adequate staffing, balanced workloads, and safe and quality care for acutely ill inpatients [2,12]. A variety of approaches including professional judgement, simple volume-based methods (such as patient-to-nurse ratios), patient prototype/classification, and timed-task approaches are proposed. Tools generally attempt to match staffing to a mean average demand or time requirement despite evidence of skewed demand distributions. Despite the large volume of publications, evidence about nurse staffing methods remains highly limited [2,12].

This situation demonstrates the need to develop an evidence-based and nurse-sensitive outcomes method upon which staffing for safety, quality, and workplace equity can be measured as well as an instrument that projects in a reliable and valid manner nurse staffing requirements in a variety of clinical settings [2], that is, a scale that can measure nursing workloads objectively and that allows the adjustment of the nursing staff to the actual demands of patient care covering the four nursing functions (patient care, teaching, management, and research). It should include standardized and internationally known nursing language not only to ensure patient safety, reduce morbidity and mortality, and increase the quality and patient satisfaction with the health care provided [4] but also to avoid burnout syndrome in nursing staff and increase their job satisfaction, which will result in the overall improvement of the health system within any health institution [13,14]. The instruments for measuring workloads in nursing are an essential tool for making decisions in the management of nursing human resources, provided that they are holistic, taking into account all the interventions carried out during their work shift, whether direct with the patient or indirect and, if possible, using standardized language that accredits that they are specific tasks of the nursing discipline [10,15,16,17]. An appropriate workload measurement tool would be designed under health care criteria that govern and guide towards the professional competencies required to be performed with the necessary resources, obtaining the expected results [18].

As we have said before, it is necessary that workload measurement instruments to be based on or include internationally standardized language so that all nurses can identify it, such as that offered by The Nursing Interventions Classification (NIC) that describes specific behaviors of the act of caring and enables comparisons between the care practiced in different scenarios; it is regularly updated [18,19,20,21], and that allows the systematic organization of care treatments performed by nurses, and an estimation of the time taken to carry out the intervention is included in its characteristics [21]. A nursing intervention refers to any treatment or care based on clinical judgment and knowledge that the nurse performs to enhance patient outcomes [19,22]. Each intervention details a series of activities, which encompass the specific behaviors or actions developed by nurses to implement an intervention to help the patient to improve his/her health outcome [19,20]. Therefore, the more complete the measurement instrument is in that it faithfully collects all the possible activities to be carried out by the patient within the four nursing functions, the more reliable the measurement will be, and the nursing staff will be able to adapt to the actual care demands of inpatients at that time.

The design and application of a measurement instrument must be preceded by its validation, which includes the vision of healthcare professionals [6] who will use it. A pilot test should be performed in real-life conditions in different hospitals before being approved for use. In terms of measuring workloads in nursing, it should measure nursing activities in each of its four functions, which allows knowledge of the activities of the discipline itself, those delegated, and those shared with other professional groups and the time spent and the number of times that each of them are carried out during the 24 h of the day throughout the different work shifts. Providing nursing care is linked to the time required to perform it and to the nurse–patient ratios to cover these times 24 h a day, 365 days a year. In this way, it will also allow to know the real situation of the level of dependency or autonomy of the patient and the need for human resources to cover all these activities or interventions to ensure proper quality management and administration of care, resulting in improved satisfaction of the recipient of nursing care and patient safety [6,22].

This research project is financially supported by The Institute of Health Carlos III (AES—Spanish acronyms for Strategic Action in Health-2018 call) [23] in relation to the Spanish National Plan for Scientific and Technical Research and Innovation 2017/2020, aimed at projects and initiatives in health services research as a research priority in the challenge “Health, demographic change and well-being”(World Health Organization, 2014) and within the “Spanish Pluriregional ERDF Operational Program (POPE) 2014–2020” (PI18/00950) [24]. Its main objective is to contribute to improving the management of nursing human resources and the quality and administration of health care to ensure patient satisfaction and safety [6,22] through the design, validation, and application of a nursing workload measurement scale for adult hospitalization units based on nursing interventions (Nursing Interventions Classification, NIC) [18,20] and, in this way, contribute to improving the quality and evidence of nursing workload measurement instruments. As we have previously developed, although the importance of nursing staffing levels in acute patient units is widely recognized within a hospital, the evidence of tools to determine the necessary human resources, even though they are extensive, is not very effective. The hypothesis that we proposed was a scale based on nursing interventions with standardized language, such as those included in The Nursing Interventions Classification (NIC) [20], and structured according to the four nursing functions (care, teaching, management, and research); it is valid and reliable to measure workloads of nursing in any adult hospitalization unit although the internal medicine and surgery units were chosen to be the most representative of any hospital. Thus, it can establish the appropriate staff for the real demands of patient care, improving the quality of care provided by these professionals. The aforementioned scale has been registered under the name MIDENF^®^ and is associated with software for data recording and analysis and is designed and registered within the same project.

## 2. Materials and Methods

### 2.1. Type of Study

This is an observational, analytical, descriptive, and prospective study, with qualitative and quantitative methodology. Documentary review and fieldwork were used as the working methodology in different versions depending on the methodology used in each phase.

### 2.2. Methods

The aim of this study, as the first stage of the project, was to design and validate a scale for measuring workloads in adult hospitalization units as a tool for measuring the activities and tasks of the nursing discipline based on the Nursing Interventions Classification (NIC) [18,20]. Then, we distributed these tasks in each of its functional dimensions, which allowed learning about the real situation of the workload and identifying the appropriate management of human resources, which ensures the quality of care [25,26,27].

An updated and rigorous bibliographic review provided the transfer of concepts from each of the nursing functions, distributing and sizing these interventions and tasks in each function (teaching, research, management, and patient care). After this previous literature review, a draft scale for measuring nursing workloads for adult hospitalization units was prepared by the research team with psychometric support. This ensures the level of validity and reliability in the selection and quantity of the sample analyzed during the validation process [10,16,17]. The scale covers and is structured based on the functional dimensions of the nursing discipline. The items were developed from a selection of nursing interventions (NIC) [20], adapting them to each of the aforementioned functional dimensions, and adapted to the tasks or activities arising from these interventions in order to be applied in adult hospitalization units.

In parallel to this study, another study was carried out in which the duration of each NIC intervention included in the scale was measured in real time. The times obtained were mapped with the standardized temporality proposed by NANDA (North American Nursing Diagnosis Association) in order to determine the time to be assigned to each item in the scale when measuring nursing workloads, which was as close as possible to the current real situation.

This scale was subjected to a validation process using a mixed methodology: qualitative (focus groups, in-depth interviews, and a part of the committee of experts) [28] and quantitative (another part of the committee of experts, piloting its application in two internal medicine units, and multicenter application in surgery and internal medicine units of four participating hospitals).

#### 2.2.1. Scale Design and Validation Process: Methodological Procedures, Sample, and Scope of the Study

##### Literature Review

Initially, a documentary study was conducted, with literature search in English and Spanish from the last 10 years (2010–2020) on the topic of study, using as search descriptors the keywords mentioned above. The search was performed on scientific databases and digital repositories and accredited websites using Boolean operators and using keywords from health sciences descriptors to create search strings.

##### Timeline and Scope of the Study

The research project started in 2018, financially supported by the Carlos III Health Institute (Spain), in its AES 2018 call, and will end in December 2022 [23].

The result of the literature review allowed to start, in June 2018, the development of a first draft of the tool, which was submitted in October 2018 to a qualitative methodology through focus groups [29]. The individual in-depth interviews were conducted from December 2018 to February 2019 [30]. From February to April 2019, the activity related to the review process by the committee of experts was carried out electronically, with the participation of relevant persons in nursing management at national and international levels, and the qualitative methodology was completed with a quantitative methodology, following the results obtained with the aforementioned methodological tool [28]. In order to carry out the focus groups and in-depth interviews, the nursing staff of a third-level hospital of reference was convened. The same hospital in which the scale obtained the qualitative validation was piloted in November 2019, and thus, its implementation in real clinical practice was tested in two internal medicine hospitalization units. After the pilot test, a multicenter application of the scale was carried out in the internal medicine and surgery units of four hospitals. It was recorded 2 days per month over 9 months in 2020.

##### Sample Selection

The participants in each part of this qualitative validation went through a random selection. The principle of saturation was used when continuing with the interviews, focus groups, or the committee of experts; that is to say, 2 focus groups, 10 in-depth interviews, and 12 experts were consulted, and considering the samples of each one of these qualitative techniques, no more contributions or new data were obtained that would justify continuing that methodology. An attempt was also made to include professionals and experts in such a way that the sample was representative of most types of hospitalization units (in terms of patients and medical specialty) that can exist in a tertiary-level hospital.

With regard to the pilot test, two internal medicine units were selected, as they had the highest number of patients and the greatest variability of nursing interventions in the hospital chosen for the pilot test. This includes the entire sample area, that is, all the patients included in these units during the period of time in which the pilot test was carried out, achieving a total of 140 scales for the pilot study during the selected measurement day in two internal medicine units during the three work shifts that day (morning, evening, and night). For the multicenter study, one scale was applied for each work shift (morning, evening, and night) to all patients admitted to the units of study during the 2 corresponding measurement days of each of the 9 months of measurement in the units of internal medicine and surgery of the four participating hospitals, obtaining a total of 11,756 completed scales. Therefore, the sample for both the pilot study and the multicenter application of the scale fully corresponded to the sample universe, that is, all the patients admitted to the measurement units on the selected days.

##### Qualitative Methodology Procedures

Two focus groups were formed, and following the methodology of this qualitative technique [29], a structured and open group interview was conducted with two representative groups of the nursing staff under study. The first group was composed of eight nurses who were references in quality health care in hospitalization units, and the second group composed of eight other professional nurses who carried out their care work in these units. Through their professional and personal experiences on the research topic, they made contributions to improve and adapt the initial design of the draft of the workload measurement scale based on NIC interventions created by the research team. Each focus group session lasted approximately 2 h and was audio and video recorded for further analysis using the specific software for qualitative methodology, Atlas.ti8 [28].

In reference to individual in-depth interviews, 10 semi-structured interviews with healthcare professionals and nursing supervisors from adult hospitalization units in a tertiary hospital were conducted to learn about their opinions, suggestions, contributions, or changes about the draft of the scale to be validated. These were audio-recorded for subsequent analysis, also using the Atlas.ti8 program [28].

After applying these qualitative methodologies, a template of the initial scale was redrafted, including the modifications suggested by the focus groups and individual in-depth interviews. This template was sent in a Word document by email to 12 national and international experts in care management and quality, who were selected from the initial bibliographic review since they were the authors of most of the selected articles, to carry out an analysis of the content validity, construct validity, understanding, and assessment of each item and of the scale in general. Each expert from the 12 consulted assessed each item of the scale individually, carrying out various types of assessment. The first two are related to the construct, one corresponds to the relevance they give to the item to be explored by using a Likert scale, and the other assessment is related to the consistency of the items. They also expressed their view on the wording and understanding of the item and what action they would like to take with each item based on the previous evaluations. At the end of each block, the qualitative methodology was included in a “comments” section requesting clarification of their decision in case of a negative assessment in each of the aspects to be assessed. There was also a section for “suggestions”, in which they could expose their proposals regarding the overall assessment and the need to incorporate new items or dimensions that were yet to be explored, analyzing in the same qualitative way as previously described [28].

##### Quantitative Methodology Procedures

The results of the committee of experts, in addition to providing a qualitative part of the validation, were also subjected to a quantitative statistical analysis that provided the level of assessment and relevance they assigned to each item and the level of coherence in the construct of the items. It also allowed us to identify the experts’ assessment on the level of understanding, the quality, and appropriateness of the wording of the items and the suitability of the time assigned to each activity in the different functional dimensions to which the activities were assigned, completing the obtained qualitative data.

A new draft of the scale was prepared, taking into account the contributions of the focus groups, individualized interviews, and arguments of the committee of experts. The pilot test began in a third-level hospital on 18 November 2019 in two internal medicine units, collecting in a sample from 140 patients the nursing activities or interventions carried out on each of them during the morning, evening, and night shifts on that day of piloting. These data were registered in a specific software for this workload measurement scale that was designed and carried out within this research project. In this way, each nurse who worked during the measured work shifts typed the web address from any computer to record in the specific created software all the interventions or activities carried out on each of their patients during that work shift.

After the pilot test, nursing workloads were measured by applying the scale in a multicenter manner in four public hospitals, some reference centers, and other regional centers in order to quantitatively validate the scale in different real-life conditions. Initially, it was planned to carry out measurements for a full year at a rate of 2 days a month in order to apply the scale in the maximum amount of different possible circumstances that may arise in a full year, but due to the pandemic situation experienced during data collection, the measurements were carried out 2 days a month for 9 months in 2020 in the internal medicine and surgery units of each participating hospital, applying the resulting scale to all patients admitted on the days of measurement in the selected units, at the rate of one scale per patient and work shift. Data recording was carried out in the same way as during the pilot study; that is, the nurses who worked the measured shifts performed computerized recording by using the designed software that can be accessed from any computer via a web address.

##### Evaluation Instruments and Variables Considered

To determine the items of the scale, the most common NIC interventions carried out in adult hospitalization units were first selected, based on a previous bibliographic review, the work experience of the research team, and the contributions of the professionals who work in these units through the results obtained in the focus groups and in-depth interviews. On the basis of these contributions, the research team determined the items by grouping those interventions on the same topic or activity in one item so that each item includes one or more NIC interventions, taking into account that the interventions that were grouped had the same time assigned to be performed. These items were validated by the committee of experts and subsequently in real practice through their application in the pilot study and in the multicenter study to check whether the designed scale reflected the reality of nursing work.

The designed scale obtained after qualitative validation and piloting, which was applied in the multicenter study, has been registered as MIDENF^®^ and is structured according to the four functional dimensions of the nursing discipline (teaching, research, management, and care). Its items are framed within these nursing functions and were developed from a selection of nursing interventions (NIC) [20], adapting them to the tasks or activities derived from the most common interventions in the adult hospitalization units of internal medicine and surgery. In addition, each item was assigned a specific execution time after a mapping between the real time measured in current care conditions and the time standardized by NANDA to approximate as much as possible to the current reality.

The MIDENF^®^ scale consists of 21 items; each item contains one or more NIC nursing interventions associated with the same application time. The scale is applied to each patient in each work shift, noting the number of times each intervention/item is performed. The total time dedicated to that patient is calculated by adding the times resulting from each intervention performed. The care workload of a nurse is calculated by adding the time dedicated to each of the patients they attend during that work shift. To this time, the time dedicated to managing the unit, teaching, and research during the same work shift, is added to determine the total workload of the nurse in the measured work shift. The MIDENF^®^ scale consists of 15 items for the care function, with their corresponding execution times: self-care (17 min), prevention (2 min), medication (9 min), samples (5 min), health education (3 min), nutrition (7 min), common invasive procedures (11 min), wounds care (9 min), fluid Therapy (22 min), care of devices (13 min), monitoring (2 min), airway (6 min), positioning (4 min), comfort (3 min), and patient and family support (8 min); 4 items for the management function: 3 items for patient management, 9 min each (which includes care provided upon admission and discharge from the unit) and 1 item for unit management, 21 min); an element for teaching (16 min); and 1 item for research (20 min). In addition, it includes a separate set of items considered as complementary, which are activities that are usually carried out on an occasional basis in these units although less frequently than the previous ones and also have their time assigned: cardiac arrest (35 min), complex administrations (chemotherapy, 18 min; blood products, 10 min), transfers (60 min), occasional invasive procedures (9 min), isolation (11 min), behavior (50 min), interventions shared with the physician (27 min), and end-of-life care (38 min).

In addition to recording the data corresponding to the number of nursing items/activity performed on each patient in each work shift on the measurement days both in the pilot study and in the multicenter study and the corresponding time invested in performing them, which provided data on the amount of workload corresponding to each function and the total for each patient and unit, sociodemographic variables were also recorded, such as age, gender, days of stay, medical specialty, admission diagnosis, etc. At a qualitative level, opinions were also collected not only about each item and its corresponding execution time, but also the need to measure workloads, how human resources management influences health care professionals and patients, their opinion about the four nursing functions, or the NIC terminology were some of the aspects, among others, that were collected in the results of this study.

##### Statistical Treatment and Data Analysis

The results of the pilot test and the multicenter study were subjected to descriptive analysis, inferential analysis, and reliability analysis using R software version 4.0.3 [31].

As the objective was to determine whether the scale had the capacity to exhibit consistent results in successive measurements of the same phenomenon, the reliability coefficient was determined. It corresponds to an index, which, in the form of a proportion, would provide information between the variance of the true score of the scale and the total variance to determine, probabilistically, the degree of variation attributable to random or causal errors not linked to the construction of the instrument. This ensures the consistency expressed in the determination of the degree of error in the application of a scale and, therefore, in the measurement of the phenomenon.

In addition, in order to check whether the instrument was detecting any statistically significant differences, a descriptive and inferential analysis of the data obtained in the initial pilot test was carried out. The descriptive analysis consisted of 41 variables, showing “hospitalization unit” and “shift analyzed” as independent variables. These variables were crossed with the dependent variables that allow calculating the time spent by nurses in those activities and tasks related to the selected nursing interventions (NICs): self-care, preventive activities, medication administration, samples collection, health education activities, nutrition, invasive procedure activities, wound care, fluid therapy, care of devices, airway care, monitoring activities, patient comfort care, family support, cardiorespiratory arrest activities, chemotherapy, transfusions, transfers to other units, occasional procedures, patient isolation interventions, patient companionship, interventions shared with the physician, care of terminally ill patients, management activities besides those abovementioned (admissions, patient discharges, administrative requirements of the unit), and research and teaching activities. All these activities were analyzed in every single unit and shift.

An inferential analysis was then performed to identify significant differences in the variables measured based on the variables “hospitalization unit” and “shift” and the interaction between the two (unit by shift). To do so, the assumptions of normality and homoscedasticity were initially tested using the Shapiro–Wilk and Fligner–Killenn tests, respectively. When the assumptions of normality were met in each of the crosses of the two independent variables (unit and shift), and the homoscedasticity assumption was met, a two-way ANOVA was used for the analysis. If any of these assumptions were not met, the Welch ADF test (robust two-way ANOVA) was used.

## 3. Results

The results of the qualitative methodology, namely the focus groups, and individual in-depth interviews, showed coincidences in the statements and opinions of the participants. These were analyzed and represented by means of the Atlas. ti8 through a network of codes assigned to each topic on which we worked. These codes were associated with a textual quotation of the participants that was representative of that code and interrelated among them, which showed how the participants valued the scale, its items, the functional dimensions involved, their personal vision of the profession and of human resources management, and the need and importance they attached to measuring workloads [28] (Figure 1).

The central codes or themes that encompass the others are: “nursing management” and “nursing workload measurement scale” in the central part of Figure 1, which are related to each other. In turn, each of them has other codes that are part of them or are associated with them. Within the “nursing management” code we included “importance of measuring workloads” and “importance of nursing in management”; that is, within nursing management, it is very important to make visible the nursing management function, which is undervalued compared to others such as care. With this code, we wished to find out what the nurses themselves think of their management capacity and the fact that the nurses themselves manage and not other professionals. In addition, within nursing management, one of the most important issues is to measure workloads adequately in order to adjust the nurse–patient ratio to the real demands of patient care, and we wished to find out if the nurses believed it is necessary to carry out these measurements to improve human resource management. In turn, the “nursing management” code is associated with two others: “impact of workloads on professional nurses” and “impact of workloads on patients”, where we wished to find out the impact of nursing management and specifically the measurement of workloads for patients and nursing professionals. Within the code “impact of workloads on professional nurses”, there is another one that we named “limitations of nursing work”; that is, what characteristics or limitations does nursing work have that can influence the impact of workloads on nurses. Within the code “impact of workloads on patients”, we included “patient satisfaction” since they are two concepts that go hand in hand and since it will affect his/her satisfaction with quality of care received depending on the impact of workloads on patient care.

The second part of Figure 1 shows the codes related to the other main theme detected: “nursing workload measurement scale”; that is, the qualitative analysis focused on the workload measurement scale itself that we are validating. This main code was associated with two others, which are: “application of the scale in real life”, which shows the opinions of the participating nurses on the application of our scale in real conditions according to their professional experience to find out if they consider it adequate to use it in their real working conditions, and the “opinion on the scale” code; that is, if they consider this scale adequate, and their opinion on the items and time allocated to them, if it is complete and includes all the real nursing activities, if they consider it easy and useful to apply, etc. In addition, “nursing workload measurement scale” has two codes or key elements that are part of it and that contributed to its creation, which are “NIC interventions” since it is designed based on internationally known NIC interventions, and each one of those that appear on the scale was described by the other code “time required to perform an intervention”, with which it was associated. In this way, we can determine what the nurses participating in the qualitative validation think about the content of the scale, its items based on NIC interventions, and the time each one takes, which will help us to measure the workloads. Associated with these last two codes, we have the codes corresponding to the four nursing functions: “care function”, “management function”, “teaching function”, and “research function”. Since the scale is structured according to these functions and includes NIC interventions of each of them, it takes into account all the work carried out by a nurse and aims to be as complete and adjusted to reality as possible.

With regard to the results of the committee of experts, we can highlight that the best-valued items in terms of relevance (91.67%) were “prevention” and “health education activities” (Table 1). The items “self-care assistance”, “prevention”, “medication”, “health education”, “nutrition management”, “care of devices”, “environmental management”, “change of position“, “comfort”, “emotional support”, and “active listening” as well as “family support”, “care of the dying patient” or “end-of-life care”, and “teaching activities” during work obtained a 100% consistency rate with the construct (Table 1). The most highly rated items for their wording were “health education” (88.33%) and 100% for “help in self-care”, “prevention”, “monitoring”, “post- or positional changes”, and “care at the end of life” or “care of dying patient” [28].

The results of the comments related to “difficulty in understanding the scale”, “NIC terminology”, and “time allocated” to each item to carry out the action focused on the need to have an instrument for measuring workloads for adult hospitalization units. This instrument should adapt the demand for care needs to the real situation of human resources to carry them out. All the actual needs in terms of nursing activities, not only health care but also management, research, and teaching, functional dimensions that are increasingly present and in which the spent nursing time should also be included in the scale [28].

The results corresponding to the pilot test were obtained after a descriptive and inferential analysis and determination of the reliability index. The descriptive analysis (Table 2) revealed a similar distribution of patients between the two units of 63 men and 77 women, with an average age from 64 to 76 years. The average length stay was 10.24 days, and the highest medical specialty, in terms of care received, was internal medicine in all shifts (73.10%). In terms of workload, the times spent were longer in the care dimension, especially in the morning shift in the two participating units. The workload of management activities was similar in the three shifts, with no significant differences between the two units analyzed. In terms of the time spent on selected activities, we found that self-care activities are more frequent in the morning shifts as well as medication, health education, invasive procedure care, wound care, comfort, and fluid therapy. Occasional procedures require more time in the evenings, and isolation processes are more frequent in one unit than the other in the mornings and evenings. The activities shared with the physician generate more workload in the morning shift, and the aid in dying patients presented the same workload in the three shifts, with one of the units having a higher workload in the evening shift. The activities of management and listening to the patient require great nursing attention in terms of time in the three shifts and in the two units studied.

With regard to the study of reliability and internal consistency, for the initial pilot test, with 140 scales completed (63 for men and 77 for women, 45% and 55%, respectively), the reliability and internal consistency of the questionnaire was calculated in a general way using different indexes in the pilot study. Cronbach’s alpha resulted in α = 0.631. According to George [32], this result is acceptable. A composite reliability index of 0.452 and an average variance extracted (AVE) index of 0.106 were also obtained, which can be improved [33]. Finally, McDonald’s omega yielded a value of 0.640, considered acceptable [34]. Afterwards, construct validity was performed to measure the latent variable “perception of work performed” by means of a factorial analysis of principal components with varimax rotation. A significant *p*-value of 0.000 was obtained for Bartlett’s test of sphericity and a Kaiser–Meyer–Olkin (KMO) coefficient of 0.520 for the proportion of the variance that the variables analyzed have in common (an acceptable sample adequacy is considered from 0.5). It was previously verified that all the variables of the scale correlated adequately, and no multicollinearity occurred. It also obtained a significant result in Bartlett’s test that corroborated that the correlation matrix was not similar to the identity matrix. Referring to the additional inferential analysis performed on the additional pilot test data, Table 3 shows the significant results obtained, for example, for the “*p*-value” of the mixed models (level of significance), from which we say that the differences are statistically significant when it is less than 0.05 in respect to the unit, the work shift, and regarding the interaction, that is, when there are significant differences regarding the unit, work shift, and regarding the interaction between them. In Table 3, the significant values have been highlighted, obtaining statistically significant values both in the unit and the shift as well as in their interaction in the items “prevention”, “care of devices”, and “monitoring”.

Furthermore, when applying the scale in a multicenter way in four hospitals in the region, 11,756 completed scales were collected, 5963 from men (50.7%) and 5793 (49.3%) from women, for 57 variables, obtaining similar results to those of the pilot test, as the highest workload was obtained in the health care function, followed by the management workload, especially in the morning shift. The aforementioned indices improved in the final application of the questionnaire. A Cronbach’s alpha of 0.727 (considered good), a composite reliability of 0.685, and an AVE of 0.234 (considered good) and an omega of 0.704, also considered good, were obtained.

## 4. Discussion

One of the most relevant qualities of our study is the fact that it uses a qualitative and quantitative methodology to validate a scale and not only one. This provides a more complete analysis of its reliability and validity. It is increasingly necessary to use both types of methodology and not just quantitative ones to check the validity or reliability of an instrument [8,35], as it complements the numerical data provided by the quantitative methodology. Likewise, the professionals who will use it in their day-to-day activities provided a qualitative vision, which represents greater value. We found some recent studies that used qualitative methodologies to validate instruments in a similar way to the study that we present. They are in line with our vision: they provide a critical analysis to improve the initially proposed instrument [8,19] although they focus only on some specific medical specialties, such as oncology units [8,19]. We also found studies that studied workloads quantitatively in patients from other specialties, such as neurology and traumatology [36] or pediatrics [37] and even in medical and surgical units [37], similar to those studied in our project of research but using indicators such as occupancy rates or hospital stay but not through a validated instrument specifically designed for these units and based on nursing interventions, as we have done in our study; even so, there is not much related literature on the measurement of nursing workloads in adult hospitalization units, and the need to further develop this topic is evident [2,9,38].

The application of a validated scale for measuring workloads in hospitalization units associated with nursing interventions (NIC) based on the opinions, experiences, and contributions of health care professionals and by means of the qualitative methodology instruments used (focus groups, interviews, committee of experts) shows an adaptation of its design to the real-life situation to be applied, as it is accepted by health care professionals, who, by seeing their work reflected in the instrument, can facilitate its applicability in practice and in their commitment to the improvement of nursing management of human resources, as demonstrated in other studies that highlight the importance of the nursing professionals in the evaluation of workloads or the creation of objective instruments aimed at this end [35,39].

The workload associated with actual nursing interventions in hospitalization units, as we have proposed in our scale, is the most reliable indicator for determining the necessary nursing staff. It is considered a relevant management tool when balancing the staff’s needs both in quantitative and qualitative terms [8,22,35]. That is why it is increasingly necessary to incorporate the standardized and internationally recognized nursing language proposed by NANDA into workload measurement instruments. NIC interventions, in which all the activities are reflected, and their use as a model for the development of nursing workload measurements are shown in other studies consulted [8,22], in which authors have also adopted the use of NIC interventions as a reference for measuring workloads in nursing teams [8,16,18,19]. Although there are more and more studies using the standardized NIC terminology to measure nursing workloads, it is essential to increase the number of reports as well as the settings and clinical context in which the Nursing Interventions Classification is used with greater quality and methodological rigor [21], and that is what we have attempted to contribute with our study, in this case, focused on the most characteristic adult hospitalization units of a hospital, such as those of internal medicine and surgery.

We would like to highlight that the mapping of interventions carried out, similar to other studies [22], as well as the actual measurement of their times compared to those standardized by NANDA for NIC interventions carried out by our research team, provides the scale a real added value and greater visibility of professional work. Together with the assessment of the four main functions of nursing, this has contributed to identify less-valued or less-visible activities that a nurse also performs, which are essential to provide quality care that has continuity in different levels of care [40]. The study of the time allocated to each intervention or nursing activity is also a poorly developed topic in the literature consulted to measure workloads in nursing since the systematic reviews consulted [21,41] show that most of the existing studies use the methodology of the minimum-required nurse–patient relationship, and very few evaluate the number of nurse hours per patient day staffing methodology, with even fewer studies based on identifying specific interventions, types of activities, the prevalence of interventions, and the time required to perform them [21,42,43], such as the study we present.

In our study, a greater workload was obtained during the morning shift, and therefore, the items or interventions that are usually carried out in the morning were the ones that presented the greatest workload quantitatively, coinciding with the fact that they are also those that obtained a 100% concordance rate with the construct, such as “self-care assistance”, “medication”, “health education”, “care of devices”, “change of position” or “comfort”, among others. In line with other studies consulted, even with different naming of the items to be measured, they also obtained more workloads in those included within the physiological domain [16]. We have to highlight that one of the limitations that we found when measuring the items of the scale is the fact that many interventions are carried out at the same time, giving rise to the phenomenon of “multitasking”, which we also found in other similar studies [22,44], which shows that sometimes, the time collected from all the interventions carried out by the nurse to their patients in a work shift is greater than the hours of that shift. In addition, by contemplating all the possible interventions in the four nursing functions and not only those included in the care function, as occurs in other workload measurement studies [40], records related to management, documentation, care and support for the family, etc., which are less studied or valued bibliographically until now, as a nursing activity are included.

Finally, we would like to emphasize that our scale has demonstrated its validity and reliability not only predictively or in ideal conditions but also when applied in real conditions through a pilot test and a multicenter study in different public hospitals, with different circumstances in terms of the type of patients, health care organization, category of hospital (reference or regional), physical or structural characteristics of the units, etc., maintaining or increasing all the statistical analyses carried out despite being applied in units of different hospitals. This is an advance compared to other scales or studies that have yet to validate their instruments in real practice [8,38], which is essential for assessing their reliability and establishing their degree of precision with greater accuracy, as it indicates their real performance when used in different circumstances, their effectiveness, and suitability for any hospital center or unit where they can be applied. If we compare the reliability and validity obtained in our scale when applied in real conditions to the one obtained by other scales previously validated to measure nursing workloads, we can see that our scale obtained a Cronbach’s alpha of 0.727, which is considered acceptable, when that obtained for the adaptation of the NAS scale to Spanish was 0.373 [45] and to Portuguese 0.36 [46]. With respect to construct validity, a KMO of 0.5 was obtained, very similar to the validation of the adaptation of the NAS to Spanish, which was 0.589 [45], and its validation for Chilean intensive care units, which was 0.69984 [47]. Although these scales were designed for intensive care units [48], they have been used on occasion for adult hospitalization units [11] without reliably demonstrating that outside of their application in intensive care, they can meet their objective of measuring loads of real nursing work in any care setting, which shows that our scale, applied in eight units of four different hospitals, is valid and reliable for any hospitalization unit for adults with different realities. This reality can be demonstrated if we highlight that, for example, in our study, using a scale adapted to hospitalization units for all interventions, the greatest workload was observed during the morning shift for the two specialties in the four hospitals, while we found other publications that, using a scale designed for intensive care units (NAS) and applied in hospitalization units, showed a higher workload during the evening shift [11], as can be seen in our results, where more items with significant differences related to the work shift were obtained, showing a greater workload during the morning shift.

Other studies consulted [4] are in line with our research. Measuring the workload of nursing staff in clinical (such as internal medicine) and surgical (surgery) hospitalization units allowed identifying the appropriate proportion of patients for each health care professional based on the real demands of care in the different work shifts. Significant differences between the workloads of clinical and surgical nurses [11,37] were found, in line with our view of the need for comparison between these units, where in this field, high patient–nurse ratios, use of point-of-care technologies, and stressful working conditions require sufficiently highly skilled nurses, with few research studies available on the perceived workload, burnout, and intention to quit among medical-surgical nurses [8]. Therefore, given the current situation of the medical and surgical hospitalization units for adults, the practical implications of our study are evident due to the clear need to measure workloads to adjust nursing in a real and objective way, as shown both in the pilot study and in the multicenter study and in a quantitative way and by professionals themselves who have spoken of, in a qualitative way, the achievement, when applying this study and its scale in practice, of improving the quality of care and patient safety and the working conditions in which health care professionals work. For adult hospitalization units or for any health service where one wishes to improve human resource management by adapting to the real demands of patient care by measuring workloads, reliable instruments aimed at specific contexts are required—in this case, nursing—that cover all its professional functions, showing the interventions they carry out on the patient, whatever type they may be, defined under a common and standardized internationally recognized language, as the nursing professionals expressed to us through the different qualitative methodologies used in this study. This will allow the professionals staff required to cover the real health care needs, ensuring quality health care and patient safety in the current hospital context.

## 5. Conclusions

The validation process of our scale at a qualitative level has clearly shown the need for an instrument to measure workloads adapted to the real demands of patient care using a holistic approach, in other words, covering all nursing functions, whether they are patient care, management, teaching, or research, in which a common international and standardized language for all professionals should be used. In this case, we used the NIC interventions with times assigned based on the real situation of current nursing care, as stated by both the experts consulted and the nursing care professionals who will implement this scale in their daily activity.

In reference to the quantitative validation, the validity and reliability of the scale both in a previous pilot test and in its multicenter application has been confirmed, obtaining acceptable parameters in all the statistical indices carried out. Therefore, we can conclude that our scale for measuring nursing workloads for adult hospitalization units is valid and reliable because it measures what it has been designed for and with minor errors compared to other studies.

With regard to future lines of research based on this study, once our scale has been validated quantitatively and qualitatively with health care professionals and in patients under real-life conditions, our line of research will continue to try to incorporate this scale into the different software used for patients’ electronic clinical recording. It can thus be fed back by such records when filling in the different items, facilitating both its registration and interpretation as well as the possibility to be used and applied in other health centers in such a way that its validity and reliability can still be confirmed in any nursing context where it is applied.

## Figures and Tables

**Figure 1 ijerph-19-15528-f001:**
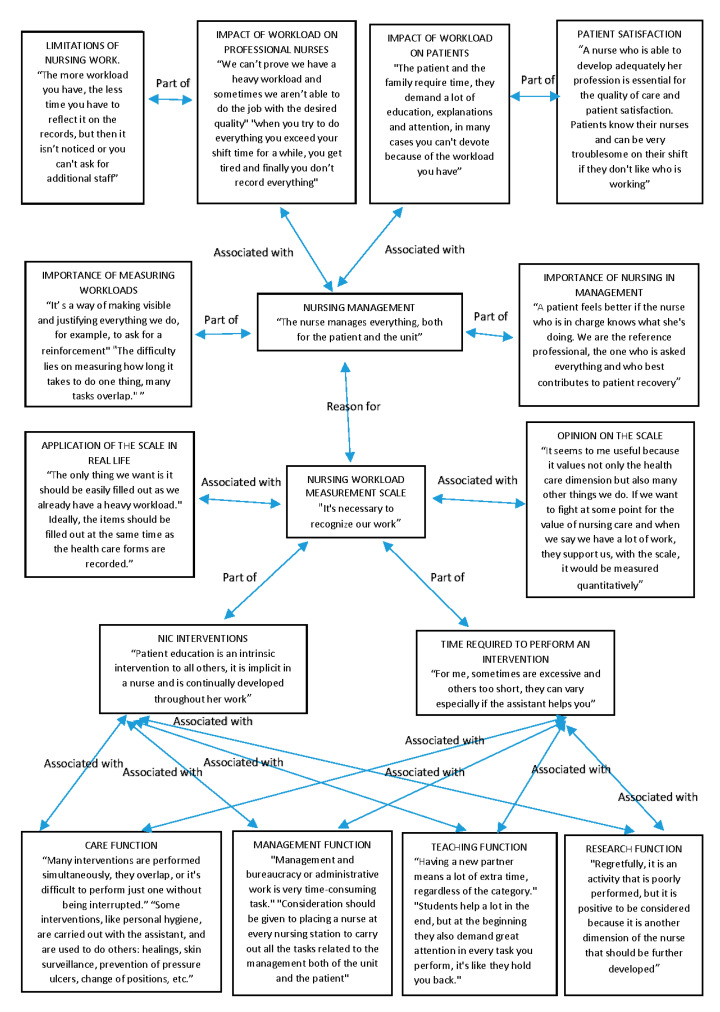
Analysis of focus groups and nursing professionals’ individual interviews [28].

**Table 1 ijerph-19-15528-t001:** Item relevance and consistency with the construct [28].

	Item Relevance		Consistency with the Construct
	High Relevance	Very Relevant	
Self-care assistance	58.3%	41.67%	100%
Prevention	91.67%	8.33%	100%
Medication	83.33%	16.67%	100%
Sample handling	16.67%	50%	91.67%
Health education	91.67%	8.33%	100%
Nutrition management	50%	41.66%	100%
Invasive procedures	75%	25%	91.67%
Wounds care	83.30%	8.33%	91.67%
Fluid therapy	75%	25%	91.67%
Airway (suction, oxygen therapy)	33.33%	66.67%	91.67%
Care of devices	50%	50%	100%
Monitoring	83.33%	16.67%	91.67%
Airway (mechanical ventilation, artificial airway)	33.33%	41.67%	83.33%
Positioning	58.33%	33.33%	100%
Cardiorespiratory arrest management	66.67%	16.67%	91.67%
Environmental management: comfort	41.67%	58.33%	100%
Emotional support, active listening	81.81%	16.67%	100%
Behavioral management	66.67%	25%	91.67%
Encouraging family involvement	41.67%	50%	100%
End-of-life care	83.33%	16.67%	100%
Patient-related management	66.67%	25%	91.67%
Unit-related management	66.67%	25%	91.67%
Teaching during work shift	58.33%	33.33%	100%
Development of clinical pathways, protocols	66.67%	16.67%	91.67%
Research data collection	58.33%	16.67%	91.67%

**Table 2 ijerph-19-15528-t002:** Characteristics of study participants.

	*n* = 140
Unit, *n* (%)	Internal medicine: 73 (52.14) General surgery: 67 (47.9)
Shift, *n* (%)	Morning: 47 (33.57) Evening: 45 (32.14) Night: 48 (34.29)
Gender, *n* (%)	Man: 63 (45) Woman: 77 (55)
Age, average (SD)	70.96 (17.74)
Days of stay, average (SD)	10.24 (10.9)
Medical specialty, *n* (%)	Endocrinology: 3 (2.14) Infectious diseases medicine: 28 (20) Internal medicine: 103 (73.57) Internal medicine 2:3 (2.14) Pneumology: 3 (2.14)
Health care workload, mean (SD)	54.34 (31.89)
Management workload, mean (SD)	30.32 (1.68)
Total workload, mean (SD)	94.49 (35.70)

**Table 3 ijerph-19-15528-t003:** Significant results of the scale variables according to unit, shift, and their interaction (unit-shift).

Variable	*p*-Value According to Unit	*p*-Value According to Shift	*p*-Interaction Value
Total workload *	0.776	**0.000**	0.135
Health care workload **	0.623	**0.000**	0.194
Self-care **	0.055	**0.000**	0.587
Prevention **	**0.000**	**0.011**	**0.000**
Samples *	0.054	**0.039**	0.338
Health education **	0.594	**0.000**	**0.002**
Usual invasive proc. **	0.647	**0.013**	0.059
Wounds care **	**0.012**	**0.011**	0.144
Fluid therapy *	0.795	**0.000**	0.837
Care of devices **	**0.030**	**0.004**	**0.021**
Monitoring **	**0.000**	**0.000**	**0.000**
Airway *	0.076	0.854	**0.008**
Position **	**0.000**	**0.019**	0.202
Comfort **	**0.000**	**0.035**	0.579
Patient and family support *	0.144	0.362	**0.035**
Occasional procedures **	**0.046**	0.139	0.139

Carried out with a two-way ANOVA (*) or a Welch ADF Test (**).

## Data Availability

The authors, responsible for the content and results presented in this article, declare: The data presented in the aforementioned article are available and without access restrictions due to the demand that may occur, always respecting the regulations governing this access to research through scientific publications. Likewise, they declare the originality of these data, a result of a research funded by the Carlos III Health Institute, and can be accessed through this article, through bibliographic references, or through the first author of this article, who, in their opinion, at the same time, is the principal researcher of the research project that generated the results presented as well as the corresponding author.

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
