# Peer review of "Validation of a Nursing Workload Measurement Scale, Based on the Classification of Nursing Interventions, for Adult Hospitalization Units"

_ijerph, 2022, doi:10.3390/ijerph192315528_

Round 1

Reviewer 1 Report

Reviewer -   Comments and Suggestions for Authors

Thank you for the opportunity to review your paper - Validation of a nursing workload measurement scale, based on the Classification of Nursing Interventions, for Adult Hospitalization Units

Overall, the content of this manuscript is interesting and the topic showed originality.

TITLE: The title is clear and appropriate.

ABSTRACT: In the abstract, the objective is clearly defined.

INTRODUCTION: introduction provide sufficient background and include all relevant references.

MATERIALS AND METHODS: The methodology used in the study is clear. The inclusion and exclusion criteria for defining the participants is clear. The number of the ethical clearance of the ethics committee was presented

RESULTS: the results are clearly presented

CONCLUSION: it is clear in the conclusion whether the study responded to the defined objectives. I suggest including contributions from the study to research and practical.

Author Response

Thank you very much for your review and comments to improve the manuscript. Regarding your suggestion to include contributions from the study to research and practice, throughout the article, the implications of our study in clinical practice are evident, the need expressed by the professionals themselves to have an instrument such as the scale that we have designed and validated with our study, as well as the advantages that it would bring both to patients, improving their quality of care and safety, as well as professionals, obtaining more adequable working conditions. Still, we have clarified or specified these implications in the discussion of the article.

Thank you very much for everything, we attach the corrected manuscript.

Kind regards.

Reviewer 2 Report

Thank you for the opportunity of available your manuscript. I suggest you the changing of the design of study, because is not a situation for a mixed study, but a methodological study, with one qualitative step and one quantitative step. The mixed design is regarding to a mixed of the results and it is does not do by the authors.

Author Response

Thank you very much for your review and comments to improve our manuscript. Regarding the option marked "Consider correct English language and style/Minor spelling required", the translator has revised the manuscript to improve the English language and correct possible errors.
Regarding his suggestion to change the design of the study, the type of design in the methodology has been modified, explained as he has indicated.
Thank you so much for everything. We attach the corrected manuscript.

Kind regards.

Reviewer 3 Report

Congratulations on an interesting research topic and an accurate methodology. A very ambitious  and needed study.

Author Response

Thank you very much for your review and for your words about our manuscript. It is an honor for us to have our article reviewed and we really appreciate that you liked it.
Thank you so much for everything. We attach the revised manuscript.

Kind regards

Reviewer 4 Report

First of all, I would like to thank you for the opportunity to read your interesting paper entitled “Validation of a nursing workload measurement scale, based on the Classification of Nursing Interventions, for Adult Hospitalization Units. I think you are tackling a timely and relevant topic that deserves attention in the scholarly debate. This is an exciting and well-conceived study for validating nursing workload measures. The authors used analytical, descriptive, prospective and observational studies, using qualitative, mixed, and quantitative methods.

Although the paper focuses on essential topic, a few concerns deserve attention. I list here to offer some suggestions for improving the manuscript.

How did you select your experts?

What process was used to generate the items based on the Nursing Interventions? What questions were asked?

How you recorded the data? How you analyzed it?

Please explain more about the face and content validity of the items. How did you know that the items were validated? We want to know the details.

How did you select the sample size for the quantitative study? Did you perform any post hoc tests for the justification of 140 participants?

The construct development process and findings are not very easy to grasp and are not clearly presented. I recommend including some tables (for results) or graphs (construct development process) in the manuscript.

The theoretical implications part is missing.

I suggest that the authors incorporate recent and more context-related articles related to variables.

I would like my recommendations to help the authors improve their work. I hope the authors will benefit from these suggestions and make the necessary amendments to strengthen the manuscript for later submission.

Author Response

Thank you very much for your review and comments to improve our manuscript. Next, we answer all the questions:

  • How did you select your experts? 

    The experts were selected based on the initial bibliographic review carried out, since many of them are authors of the published articles selected for this study.

  • What process was used to generate the items based on the Nursing Interventions? 

    To determine the items that make up the scale, the most common NIC interventions carried out in Adult Hospitalization Units were first selected, based on a previous bibliographic review, the work experience of the research team and the contributions of the professionals who work in these units through the results obtained in the focus groups and in-depth interviews. On the basis of these contributions, the research team determined the items by grouping those interventions on the same topic or activity in one item, so that each item includes one or more NIC interventions, taking into account that the interventions that were grouped had the same time assigned to be performed. These items were validated by the committee of experts and later in real practice through their application in the pilot study and in the multicenter study, to check whether the designed scale  reflected the reality of nursing work.

    This explanation has been included in the methodology section of the article.

  • What questions were asked?

    The main topics or questions that were addressed during the scale design process are shown in Figure 1 of the results, where they are accompanied by a real textual quote, the most representative on each topic, provided by the participants. With the answers, opinions and suggestions related to these topics that the participants of the focus groups and in-depth interviews provided (all professionals who work daily with patients in Adult Hospitalization Units), we designed the scale, so that thus, it better adjusts to the current healthcare reality.

    During the focus groups, in-depth interviews and expert committee, the participating professionals were asked about each of the selected items, as well as the NIC interventions that they include and the time allocated.

  • How you recorded the data?

    The methodology explains for each process carried out how the data has been collected:

    - Focus groups: "Each focus group session lasted approximately 2 hours and was audio and video recorded for further analysis using the specific software for qualitative methodology Atlas.ti8"

    - In-depth interviews: "These were audio-recorded for subsequent analysis, also using the Atlas.ti8"

    - Committee of experts: The article expresses how they were collected and what type of data was collected: "with a template to carry out an analysis of the content validity, construct validity, understanding and assessment of each item and of the scale in general Each expert from the 12 consulted assessed each item of the scale individually, carrying out various types of assessment.The first two are related to the construct, one corresponds to the relevance they give to the item to be explored, using a Likert-scale. , and the other assessment is related to the consistency of the items.They also expressed their view on the wording and understanding of the item, and what action they would like to take with each item based on the previous evaluations. At the end of each block, the qualitative methodology was included in a "comments" section, requesting clarification of their decision in case of a negative assessment in each of the aspects to be assessed. ", where they could expose their proposals regarding the overall assessment, and the need to incorporate new items or dimensions that were yet to be explored, analyzing in the same qualitative way as previously described". We added the way to collect them  and completed in the template described in a word document that was sent to the experts by email. We also  added how the experts who formed the committee were selected.

    - Pilot study: The data collected in the pilot study were collected, as for the multicenter study, as described in the article: "These were registered in a specific software for this workload measurement scale, designed and carried out within this research project". In this research project, software included in a web page was designed, which could be accessed by the nurses who worked during each work shift that was measured, and directly record the interventions or activities carried out on each patient during each shift. all computerized. The data registration website address is: https://escalaenfermeria.imib.es/

    This explanation has been included in the methodology of the article both in the section on piloting and in the section on the multicenter study.

  • How you analyzed it? The "Statistical treatment and data analysis" section explains the program used to perform the data analysis, the types of analysis and tests that are performed for each variable, as well as why this type of analysis was chosen to obtain the results of the proposed objectives. In addition, everything has been completed and referenced with bibliography when the results obtained from these analyzes are presented in the "results" section.
  • Please explain more about the face and content validity of the items. How did you know that the items were validated? We want to know the details.

    A content validation (validation of the items) was performed by health care professionals and experts, as explained in the article, and then the questionnaire was validated for reliability and internal consistency in a general way by means of different indexes in the pilot study. Cronbach's alpha resulted in α = 0.631. According to George [32] this result is acceptable. A Composite Reliability index of 0.452 and an Average Variance Extracted (AVE) index of 0.106 were also obtained, considered to be improved [33]. Finally, McDonald's Omega obtaied a value of 0.640, considered acceptable [34]. Next, construct validity was performed to measure the latent variable "Perception of work performed" by means of a principal components factor analysis with varimax rotation. A significant p-value of .000 was obtained for Bartlett's test of sphericity and a Kai-ser-Meyer-Olkin (KMO) coefficient of 0.520 for the proportion of the variance that the variables analyzed have in common (a good sample adequacy is considered from 0.5).

    The indixes indicated above improved in the final application of the questionnaire. A Cronbach's Alpha of 0.727 (considered good), a Composite Reliability of 0.685 and an AVE of 0.234 (considered good) and an Omega of 0.704, also considered good, were obtained.

    All this explanation has been included in the article.

  • How did you select the sample size for the quantitative study? Did you perform any post hoc tests for the justification of 140 participants? The pilot sample was considered adequate because by measuring all the patients admitted (sample universe) in the two selected hospitalization units for one day, a much higher sample was obtained than the number of initial items contained in the scale designed for its validation. In addition, the number of patients, their characteristics, the workloads carried out on that day of measurement, the number of professionals, etc. of the pilot study was representative of the study.
  • The construct development process and findings are not very easy to grasp and are not clearly presented. I recommend including some tables (for results) or graphs (construct development process) in the manuscript.

    A table with the data obtained regarding the relevance of each item and its consistency with the construct has been included in the results (Table 1).

    The process on how the construction of the scale was developed has been described exhaustively throughout the manuscript, especially in the methodology section, exposing the way each pase was carried out, the type of methodology followed in each phase and how it has evolved to the final scale applied in the multicenter study.

  • The theoretical implications part is missing. Throughout the article, the implications of our study in clinical practice are evident, the need expressed by the professionals themselves to have an instrument such as the scale that we have designed and validated with our study, as well as the advantages that it would bring both to patients, improving their quality of care and safety, as well as professionals, obtaining more adequate working conditions. Still, we have clarified or specified these implications in the discussion of the article.
  • I suggest that the authors incorporate recent and more context-related articles related to variables. Three recent articles on studies carried out in hospitalization units, such as ours, and another that analyzes one of the phenomena detected when carrying out our study and collecting the data, such as "multitasking", have been included in the bibliography. The bibliography related to the statistical analyzes of the data has also been completed. In total, 6 new bibliographical references have been included in the manuscript.

Regarding the option marked: "English language and style are fine/minor spell check required", the article has been revised by the translator to improve English language and correct possible errors.

In addition, all sections of the manuscript have been improved following their recommendations and those of the rest of the reviewers. We attach the revised manuscript.
We hope that we have correctly answered your questions, clarified all doubts, and that the revised manuscript is to your liking.

Kind regards.

Round 2

Reviewer 4 Report

Dear Authors, I carefully re-evaluated your paper, finding it substantially improved with respect to the version. The revised version is much better organized and has higher scientific quality. Therefore, I recommended it for publication. Thank you